# Concurrent transmission of Zika virus during the 2023 dengue outbreak in Dhaka, Bangladesh

**Anamul Hasan** [1], **Md. Mobarok Hossain** [1], **Md Fahad Zamil** [1], **Afrida Tabassum Trina** [1], **Mohammad Sharif Hossain** [1], **Asifa Kumkum** [2], **Sajia Afreen** [1], **Dilruba Ahmed** [2], **Mustafizur Rahman** [1], **Mohammad Shafiul Alam** [1] *

**1** Infectious Diseases Division, International Centre for Diarrheal Disease Research, Bangladesh (icddr,b), Dhaka, Bangladesh, **2** Clinical and Diagnostic Services, International Centre for Diarrheal Disease Research, Bangladesh (icddr,b), Dhaka, Bangladesh

* shafiul@icddrb.org

**Data Availability Statement:** All complete genome sequences were submitted to GenBank under accession PQ203661 (hZikaV/Bangladesh/icddrb-104/2023), PQ129494 (hZikaV/Bangladesh/icddrb-

## Abstract

### Background

During the 2023-dengue outbreak in Bangladesh, a diagnostic evaluation study was conducted to investigate concurrent Zika virus (ZIKV) and dengue virus (DENV) transmission in Dhaka in 2023.

### Aims

The study explored to simultaneously detect the presence of ZIKV, DENV, and/or CHIKV while considering relevant clinical and epidemiological risk factors, using a real-time multiplex RT-PCR system. Following this, it was planned to sequence the selected samples to identify genetic variations of the ZIKV infections within the population.

### Methods

This study was designed as a diagnostic evaluation, where participants meeting the inclusion criteria were prospectively recruited with written informed consent. A total of 399 febrile individuals were screened, with 185 meeting the inclusion criteria of having a fever onset within 2–5 days, along with one of the following clinical features, e.g. headache, myalgia, arthralgia or bone pain, rash, nausea, vomiting, or diarrhea and 152 undergoing real-time RT-PCR testing.

### Results

Five ZIKV-positive cases were identified, including one DENV-ZIKV co-infection. Phylogenetic analysis revealed the ZIKV strains were part of the Asian lineage, closely related to Cambodian and Chinese strains from 2019. All ZIKV-positive cases were male, residing within a one-kilometer radius, with no prior travel history, suggesting community-level transmission.

044/2023) and PQ129495 (hZikaV/Bangladesh/icddrb-147/2023). All complete metadata set is available upon request from the Senior Manager, Research Administration of icddr,b (contact via Shiblee Sayeed, shiblee_s@icddrb.org).

**Funding:** The author(s) received no specific funding for this work.

**Competing interests:** The authors have declared that no competing interests exist.

## Conclusion

This study marks the first identification of ZIKV in Dhaka city and the first report of ZIKV-DENV co-infection in Bangladesh that highlights the diagnostic challenges posed by the symptomatic similarities between ZIKV and other arboviruses and underscores the need for enhanced surveillance and public health interventions to mitigate the spread and impact of ZIKV in dengue-endemic regions.

## Author summary

This study explores the co-transmission of Zika (ZIKV) and dengue (DENV) during the 2023 dengue outbreak in Bangladesh, shedding light on important public health and epidemiological issues. While Zika is typically a mild illness for most people, it can have serious neurological consequences, such as microcephaly in infants. Dengue, another mosquito-borne virus common in tropical regions like Bangladesh, affects a significant portion of the population. The research stands out because it marks the first recorded Zika cases in Dhaka and the first instance of Zika-Dengue co-infection in the country. This discovery is crucial for both local and global health communities, as it highlights the challenge of managing outbreaks of similar viruses at the same time, complicating diagnosis and treatment. The study calls for stronger surveillance and better public health measures to manage these viruses, particularly in densely populated areas where mosquito-borne illnesses spread rapidly. This work focuses on the importance of raising awareness and enhancing disease management practices to reduce the risks posed by these infections. The overlap in symptoms and the potentially serious outcomes for vulnerable groups, like pregnant women and newborns, make understanding Zika's transmission and genetic variations critical. These insights can guide future efforts in vaccine development and public health interventions at the community level.

## Introduction

Zika virus (ZIKV) belongs to the enveloped *Orthoflavivirus* genus, with a size of approximately 11 kb positive-sense single-stranded RNA genome with 2 flanking non-coding regions along with a long open reading frame that encodes a single polyprotein [1,2]. ZIKV was initially isolated in the Zika forest of Uganda in 1947 within the mammal monkey followed by the subsequent isolation of ZIKV from a pool of *Aedes* mosquitoes in the same forest next year [3,4]. Since the first case of ZIKV infection in human beings having been reported in 1954 in Nigeria, a number of human infections due to sporadic outbreaks have been reported in Africa and Asia till 2007 [5,6]. The first major ZIKV outbreak took place in 2007 on the Yap Island in the Federated States of Micronesia, affecting approximately three-fourth of the population within a four-month period [7]. In the ensuing years of 2013 and 2014, ZIKV epidemics were reported a few Pacific Islands of Oceania [8]. In early 2015, a massive outbreak swept through Brazil that subsequently and rapidly spread throughout the South America, the Caribbean nations and other parts of the world [8–11]. According to the World Health Organization, there were total 92 countries and territories reported with current or previous ZIKV transmission [12].

Serological surveillance indicated that ZIKV circulated at low but consistent levels in Asian territories between 1952 and 1997 [13]. Its first possible interhuman urban transmission in Asia was reported in Malaysia in 1966 [14]. However, the first confirmed human case was

announced over 5-decades later in 2010 in Cambodia [15]. Based on different epidemiological data published between 2016 and 2018, ZIKV outbreaks were reported in Thailand, Singapore, Vietnam, Philippines, and India [16,17]. A retrospective sero-surveillance study by the Institute of Epidemiology, Disease Control & Research (IEDCR) in 2016 revealed one confirmed ZIKV-positive case for the first time in Bangladesh. The sample was collected in 2014 from a 65-year old male with no prior history of travelling abroad [18]. Since 2016, there was no evidence of systematic sero-surveillance on ZIKV outbreak reported in Bangladesh, to the best of our knowledge.

ZIKV infection may cause serious fetal complications, such as microcephaly, leading to adverse outcomes. The virus has also been linked to neurological conditions, like Guillain-Barré syndrome (GBS), in infants and adults [19,20]. There are currently no vaccines or medications that prevent ZIKV infection; however several potential ZIKV vaccines have shown promising results in clinical trials involving human participants [21]. In spite of such public health implication, ZIKV infection remains underdiagnosed and underreporting in Bangladesh due to its symptomatic similarities to other two endemic arboviruses- dengue (DENV) and chikungunya (CHIKV) [1,22,23]. Bangladesh faced a sharp increase in dengue cases in 2023, with 321,179 reported infections and 1,705 deaths, resulting in a 0.5% case fatality rate [24]. In our study, we intended to diagnose simultaneously the presence of ZIKV, DENV and/or CHIKV in combination with corresponding clinical and epidemiological risk factors using real-time multiplex RT-PCR system, followed by sequencing to determine the genetic variations of the arboviral infection in the population.

## Materials and methods

### Ethics statement

The institutional review board of the International Centre for Diarrheal Disease and Research, Bangladesh (icddr,b) reviewed and approved the protocols (Protocol no: 23080, Version: 1.4) for participant enrollment, sample collection and testing. The protocol was strictly followed to protect the privacy, anonymity and rights of all participants. The informed written consent was obtained from the adult participants. For children between 5 and 11 years old, written informed consent was obtained from a parent or legal guardian. For children aged 11 to 17 years, verbal assent was obtained as well in addition to written consent from a parent or legal guardian. Participants or their parents/legal guardians were contacted via telephone by the assigned study physician who conducted follow-ups to ensure the best clinical correlation for particular cases.

### Study design and participants

This was a diagnostic evaluation study with prospective recruitment of participants who met the inclusion criteria of having a fever onset within 2–5 days, along with one of the following clinical features, e.g. headache, myalgia, arthralgia or bone pain, rash, nausea, vomiting, or diarrhea. The study was conducted in the metropolitan region of Dhaka, Bangladesh. Patients coming to the diagnostic facility of icddr,b at Mohakhali, Dhaka for dengue tests were considered for primary screening. We screened and enrolled suspected individuals aged between 5 and 65 years. Anyone beyond this age limits, those who were severely ill, or those with acute conditions that suggested avoiding invasive sample collection were excluded from the study.

### Study duration and sample collection

This study was conducted at the Emerging Infections & Parasitology Laboratory (EIPL) of icddr,b between July and December in 2023. By maintaining every aseptic and universal

precaution, approximately 3 ml of venous blood was collected in serum-separator tube (SST) from each febrile patient included. Serum samples was separated by centrifugation from SSTs and subsequently aliquoted in 1.5ml Eppendorf tubes. Each sample underwent primary screening for Dengue NS1 rapid diagnostic test (RDT) using Bioline Dengue NS1 antigen kit (Abbott, Ingbert, Germany, Cat# 11FK50).

The viral RNA was extracted from 140μL of serum with QIAmp Viral RNA Mini kit (Qiagen, Hilden, Germany, Cat# 52906), according to the manufacturer's instruction. Each RNA sample was cryo-preserved in -80˚C until the batchwise real-time RT-PCR was carried out.

## Detection of ZIKV using real-time RT-PCR

Arboviral RNA (DENV, CHIKV or ZIKV) was detected simultaneously using the VIASURE Zika, Dengue & Chikungunya Real Time RT-PCR Detection Kit (CerTest Biotec, Zaragoza, Spain, Cat# VS-ZDC106L), according to manufacturer instructions. The following thermal cycling parameters were used in CFX-96 real-time PCR detection system (Bio-Rad, Hercules, CA, USA): reverse transcription (RT) at 45˚C for 15 min followed by RT inactivation and initial denaturation at 95˚C for 2 min, then 45 cycles of 95˚C for 10 sec, and 60˚C for 50 sec. Fluorescence data were collected during the extension step at 60˚C through the FAM (DENV), ROX (CHIKV), Cy5 (ZIKV) and HEX channel (Internal control). Post-PCR analysis was performed using a linear scale. Baseline thresholds were set manually on each run. Amplification curve with cycle threshold (Ct) <40 was evaluated as positive and Ct values >40 as negative. The amplification of the internal control (IC) ensures that PCR reaction inhibition can be ruled out. However, detecting the IC is not essential, as a high copy number target can lead to the preferential amplification of target-specific nucleic acids.

## Whole genome sequencing

We considered the three Zika strains with Ct values less than 32, ensuring sufficient viral load for sequencing. The Nanopore sequencing kit SQK-NBD 114.24 (Oxford Nanopore Technologies, United Kingdom) was used to produce sequencing libraries for three strains, according to the manufacturer's amplicon sequencing protocol. The libraries were sequenced on MinION MK1C device using the MinKNOW platform (72 h). Fast5 data files were base-called using the guppy basecaller v6.0.1 [25] and demultiplexed using qcat v1.1.0.(https://github.com/nanoporetech/qcat).

## Quality check and assembly

Raw fastq reads of all zika sequences were quality-checked using NanoPLot and trimmed using nanoq [26] if necessary. Passed Reads were mapped against the reference genome (NC_012532.1) using minimap2 [27] and then SNPs were identified using SAMtools (v1.10) [28]. SNPs were discarded if they had strand bias $p < 0.001$, mapping bias $p < 0.001$ or tail bias $p < 0.001$ (using vcfutils.pl script, from SAMtools). Finally, to create the consensus genome ivar (v1.4.3) was used [29].

## Phylogenetic tree construction

At first a global phylogenetic tree was constructed by taking all complete sequences of zika viruses from GISAID. After taking 89 samples, including the study sample (n = 3), a representative sequence was randomly selected from each node. Alignment was done based on the E gene using augur (v24.4.0) [30] by considering NC_012532.1 as a reference genome. Maximum likelihood tree (MLT) was constructed based on the E gene alignment using IQ-TREE

(v2.0.6) [31]. Support for the MLT was calculated using 1000 bootstrap pseudo-analyses of the alignment.

## Result

Initially, a screening procedure was conducted on 399 individuals. Among them, 185 met the inclusion criteria and gave written informed consent/assent, thus enrolling as study participants (Fig 1). Regardless of their Dengue NS1 RDT findings, RT-PCR was performed on 152 samples, resulting in 50 as tested PCR-positive for DENV and the remaining as negative. Among the 50 DENV-positive samples, one was also positive for ZIKV, indicating a DENV-ZIKV co-infection. Rest of 102 DENV-negative samples, four tested positive for ZIKV. Therefore, a total of five ZIKV-positive samples (3.3%) were identified out of the 152 PCR samples.

### Clinical and demographic observations

All five identified cases were male, lived within a one-kilometer radius of each other, and had no travel history outside the country in the past two years. Their enrollment indicated that the cases were spread over a little more than a month, suggesting ongoing transmission during that period. The median age of the cases was 24 years. Headache was the only symptom reported by all participants, though other symptoms varied among the cases. The occupations of the patients are split between service workers (n = 3) and students (n = 2), indicating different social and professional settings. The recorded body temperatures were slightly elevated at the time of enrollment suggesting mild febrile illness (Table 1).

### Findings of real-time RT-PCR analysis

All five cases tested negative for the Dengue NS1 antigen using the RDT. RT-PCR results were consistent with the NS1 RDT for all cases, except Case-4. Lower Ct values are indicative of

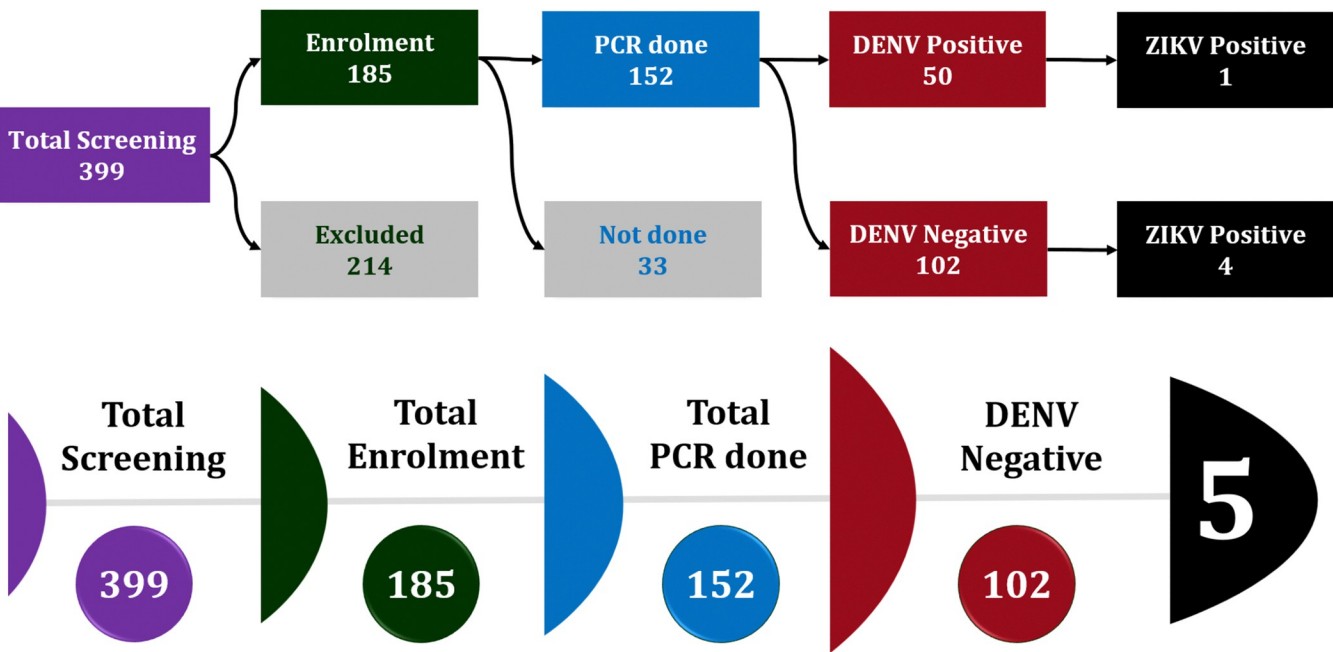

**Fig 1. Screening and diagnostic workflow highlighting ZIKV detection among DENV-positive and DENV-negative cases.** The "Not Done" category represents enrolled cases whose RT-PCR was not conducted.

**Table 1. Clinico-demographic features of ZIKV positive cases (n = 5) at the time of enrolment.**

| Variables | Case-1 | Case-2 | Case-3 | Case-4 | Case-5 |
|---|---|---|---|---|---|
| Gender | Male | Male | Male | Male | Male |
| Age range (Years) | 21–25 | 41–45 | 21–25 | 26–30 | 21–25 |
| Occupation | Service | Service | Student | Service | Student |
| Travel history in abroad in last two years | No | No | No | No | No |
| Duration of fever (Days) | 2 | 3 | 2 | 3 | 4 |
| Body temperature (˚F) | 99.8 | 99.2 | 99.1 | 98.9 | 98.5 |
| Signs & symptoms | • Headache<br>• Bone & joint pain<br>• Sore throat | • Headache<br>• Bone & Joint pain<br>• Nausea & vomiting<br>• Weakness | • Rash<br>• Headache<br>• Muscle pain<br>• Nausea & vomiting | • Rash<br>• Headache<br>• Muscle Pain | • Headache<br>• Muscle pain<br>• Pain behind the eye<br>• Weakness |

higher viral loads, with Case-1 suggesting the highest ZIKV viral load (Ct value of 25.51) and Case-2 the lowest (Ct value of 36.21). However, Case-4 tested positive for DENV via RT-PCR with a Ct value of 29.24, indicating a moderate DENV viral load. This case was also co-infected with ZIKV, with a Ct value of 34.16 (Fig 2).

## Phylogenetic tree analysis

A phylogenetic tree was constructed based on the 89 E gene sequences including 3 Bangladeshi ZIKV strains from this study (Fig 3). They were Case-1 (hZikaV/Bangladesh/icddrb-104/2023), Case-3 (hZikaV/Bangladesh/icddrb-147/2023) and Case-5 (hZikaV/Bangladesh/icddrb-044/2023). The ZIKV dataset used in this study included sequences representing two distinct lineages, African and Asian, which formed separate clusters. Furthermore, the Asian lineage formed two separate clusters of the Asian and South American sub-lineages. All Bangladeshi ZIKV strains were closely related to each other and belonged to the Asian sublineage. They formed a distinct node that was clustered with Cambodian and Chinese ZIKV strains from 2019. Sequencing analysis of ZIKV isolates from Bangladesh revealed several insertions. At reference positions 72, 137 and 7158, all three isolates (ZikaV/Bangladesh/icddrb-001/2023, ZikaV/Bangladesh/icddrb-002/2023, and ZikaV/Bangladesh/icddrb-003/2023) exhibited a 1 bp insertion. At position 1432, different 12 bp insertions were observed: isolate 001 had a "TTAATGAC" insertion, isolate 002 had a "GTTAATGACACA" insertion, and isolate 003 had a "TTAATGACACA" insertion. Additional 1 bp insertions were detected at positions 4981 and 6526 in isolate 003, and at position 8030 in isolate 002. Finally amino acid substitution analysis revealed that ET156del, V153del were present in hZikaV/Bangladesh/icddrb-044/2023 and N163D, V473M were presented in both ZikaV/Bangladesh/icddrb-104/2023 and ZikaV/Bangladesh/icddrb-147/2023.

## Discussion

To our knowledge, this was the first cluster identification of Zika-infected patients residing in Dhaka, the capital city of Bangladesh. Besides, in terms of co-infection, we reported the first case of DENV-ZIKV co-infection from Bangladesh as well. There are two main points of concern regarding ZIKV infection. First, three ZIKV strains we detected phylogenetically belongs to be of the Asian lineage. Second, none of the five suspected cases had any history of international travel in the last two years from study enrollment.

The strain analyzed in our study belonged to the Asian lineage.. The ZIKV strain from the Asian lineage was responsible for microcephaly and other neurological disorders in the Pacific regions and the Americas resulting in increased viral pathogenicity [20,32–34]. It was

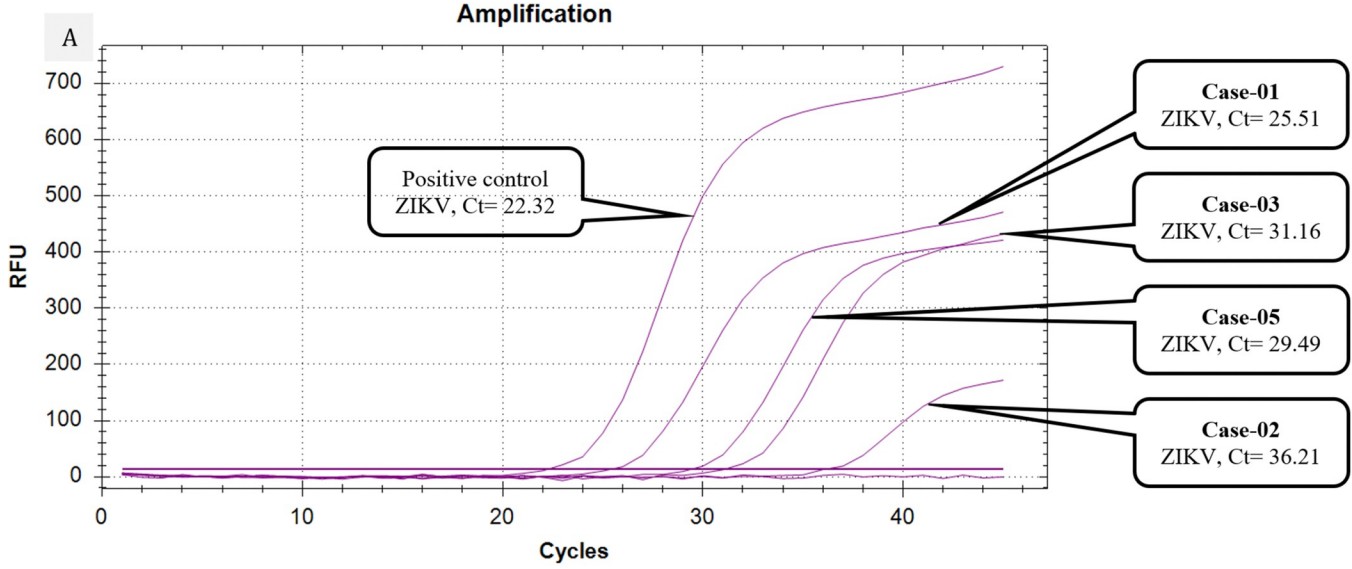

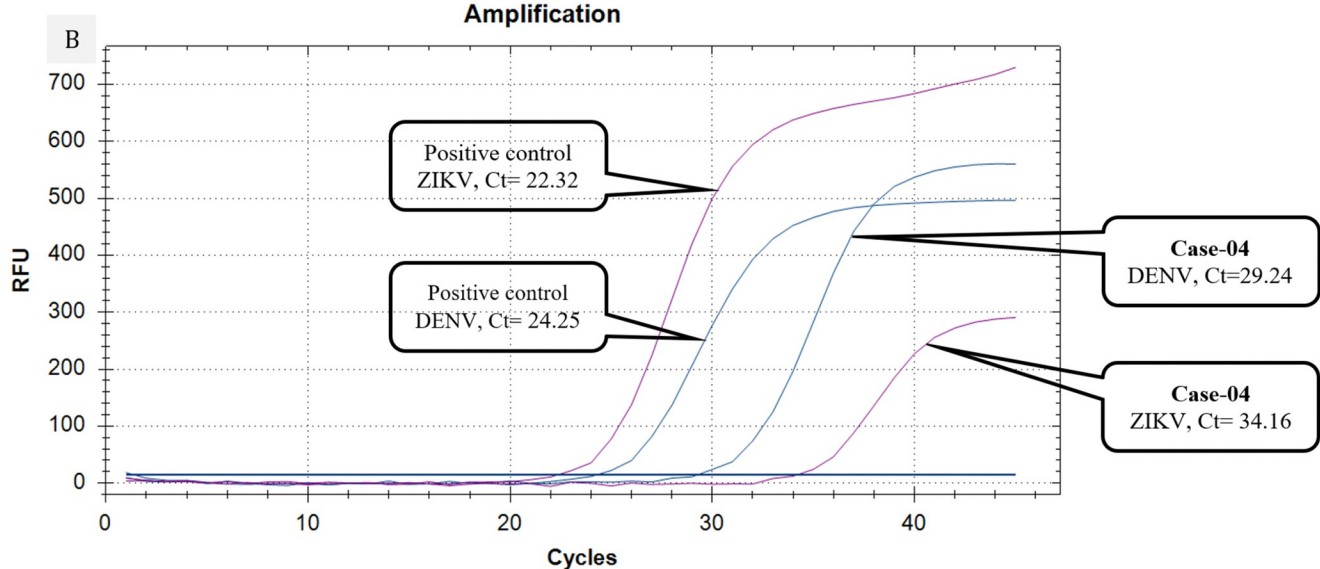

**Fig 2. Technical performance of the probe-based real time RT-PCR.** A) Amplification curves of four cases with ZIKV mono-infection have been depicted with corresponding positive control of ZIKV, each containing different viral loads with corresponding threshold cycle (Ct) values; B) amplification curves of Case-4 with co-infection have been depicted by two different colors with corresponding positive controls of DENV and ZIKV.

recognized as an emerging virus after the Asian lineage caused a major outbreak in 2007 in the Federated States of Micronesia [35]. This lineage subsequently spread to Brazil and other countries and territories in the Americas through the French Polynesia and other South Pacific islands [8,36].

Transmission dynamics of ZIKV can be mosquito-borne [37,38], sexual intercourse [39,40], blood transfusion [41], mother-to-child horizontal transmission [42] or even by secondary non-sexual physical contact [43]. So, the absence of international travel among the suspected cases residing within a 1-km radius of each other conferred a sustained community-level transmission that could have grave consequences in future Bangladesh, mirroring

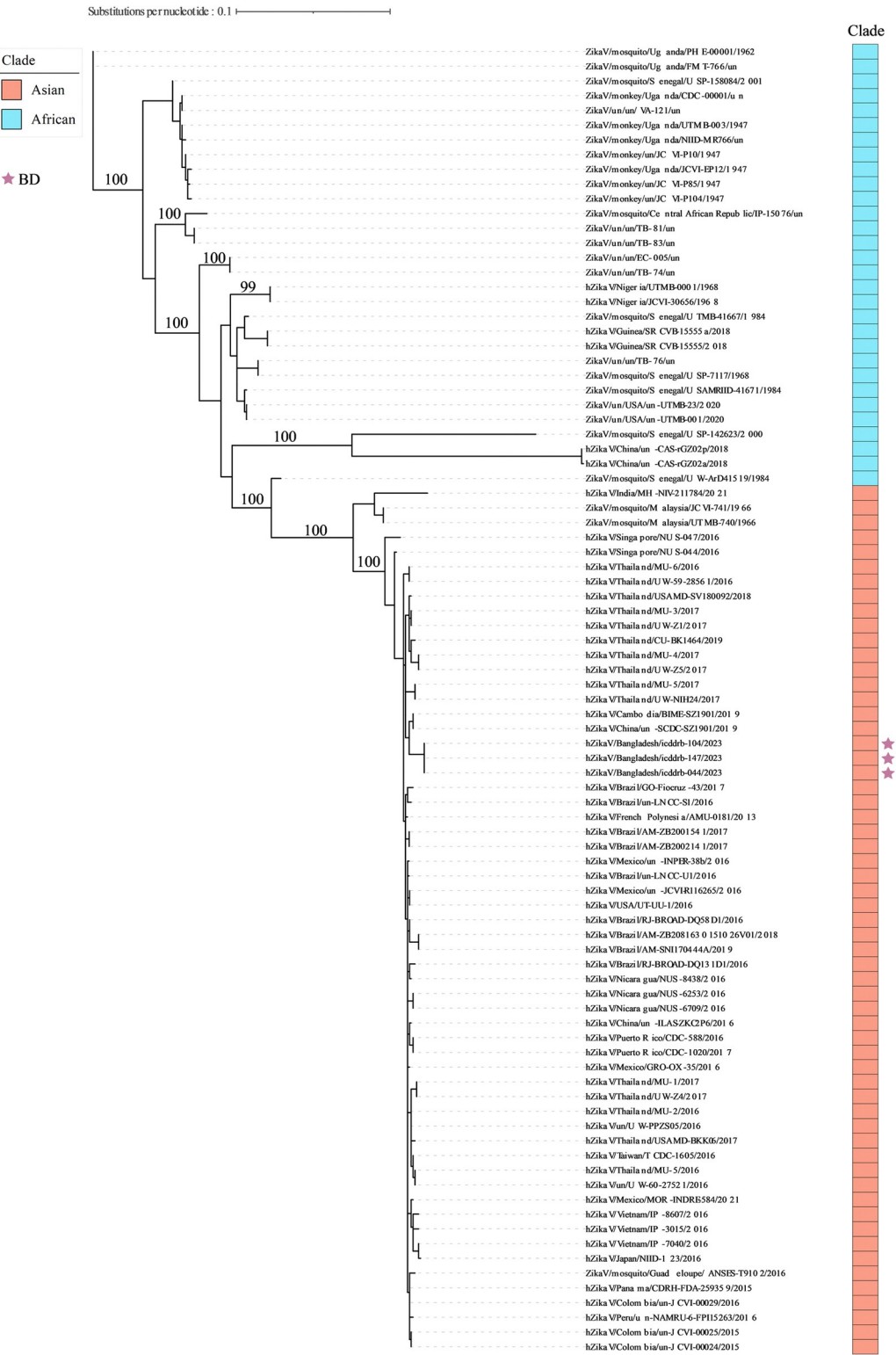

**Fig 3. The maximum likelihood phylogenetic tree of 89 E gene coding sequences of ZIKV circulating worldwide.** The tree was constructed using the bestfit substitution model with an ultrafast bootstrap of 1000 replicates. The Bangladeshi (BD) strains are denoted by purple asteroids. All three BD ZIKV strains were closely related to each other and classified under the Asian sublineage. They formed a distinct node, grouping together with Cambodian and Chinese strains from 2019.

previous study conducted in Bangladesh [18]. A lot of Bangladeshi immigrants working in different ZIKV-affected countries (e.g. Singapore, Malaysia, Thailand) may augment the possibility of spreading ZIKV infection within communities. This is because many returning immigrants prioritize family planning, potentially facilitating the horizontal transmission of ZIKV to future generations [18,44].

As all suspected cases were male with an average age of 24 years, the probability of male-to-female ZIKV transmission and subsequent pre- and peri-natal transmission from mother to descendants would be higher, potentially resulting in the birth of microcephalic neonates [4,18,45]. The causal link to severe neurodevelopmental defects like microcephaly was one of the major reasons behind the WHO's declaration of ZIKV as a Public Health Emergency of International Concern (PHEIC) on February 1, 2016 [8,10]. Rarely have fatal cases of ZIKV been reported in otherwise healthy individuals [38]; but its associated morbidity and mortality have been reported with other comorbidities, such as hereditary sickle cell anemia [46] and Guillain-Barré syndrome (GBS) [19,33].

It took approximately 8-long years for ZIKV infection to get reported and resurfaced in Bangladesh. Such an approach towards ZIKV, in contrast to DENV cases, could be because many ZIKV infections were subclinical and did not necessitate medical attention. Approximately 20% of individuals infected with ZIKV develop a noticeable febrile illness [38]. Therefore, a potential ZIKV transmission might have been obscured by simultaneous periodic outbreaks of DENV in recent years in Bangladesh [47]. For example, Bangladesh endured its most extensive and deadliest dengue outbreak in 2023 since the virus resurfaced in the country two decades ago [24].

In our study, we observed a case of DENV-ZIKV co-infection (Case 4) that could be alarming due to their synergistic effect of antibody-dependent enhancement (ADE) [48]. DENV-mediated ADE not only increases the likelihood of severe disease but also facilitates the persistence and proliferation of ZIKV in DENV-endemic regions [24]. This situation is particularly concerning as it indicates a mutualistic relationship between DENV and ZIKV within the urban transmission cycle of *Ae. aegypti*, combining pathways of viral replication and antiviral suppression, exhibiting distinct vector competence phenotypes, and resulting in increased susceptibility and transmissibility [49]. Finally, the negative results for DENV NS1 RDT in all cases point out that RDT may not be sufficient for diagnosing dengue in the context of co-infections or situation with low viral loads. The negative results for DENV NS1 RDT in all cases indicate that RDT might be inadequate for diagnosing dengue, particularly in scenarios involving co-infections or low viral loads.

There are few limitations that need to be addressed. This study was not any epidemiological nationwide survey as we could not find all zika cases by surveillance work. we did not analyze the risk of hospitalization and the occurrence of hemorrhagic fever associated with secondary infections.

According to Communicable Diseases (Prevention, Control and Eradication) Act, 2018, Zika is a public health notifiable disease in Bangladesh [50]. The identification of Zika cases was immediately notified to the designated persons at the Directorate General of Health Services of Bangladesh via email, along with detailed patient information and their consent. The corresponding IRB of icddr,b was notified to obtain permission to inform the patients about our findings as well.

## Conclusion

This study emphasizes an urgent call for enhanced surveillance and diagnostic initiatives to address the public health challenges posed by this emerging arbovirus- ZIKV. Strengthening

molecular diagnostic capacities and integrating ZIKV testing into routine arboviral surveillance systems are critical steps forward. Additionally, targeted public health campaigns focusing on ZIKV prevention and community education should be implemented to mitigate potential risks. Continued research is essential to better understand the local epidemiology and transmission dynamics of ZIKV, ultimately guiding effective interventions and policy strategies in Bangladesh.

## Ethics statement

The study was approved by the institutional ethical review committee (ERC) of the International Center for Diarrheal Disease Research, Bangladesh (icddr,b). The institutional review board (IRB) approved the study with protocol no PR-23080. The corresponding ethical consent/assent has been obtained from the study participants.

## Acknowledgments

We are grateful to the Governments of Bangladesh and Canada for providing core/unrestricted support to icddr,b. Our special gratitude goes to the field study team for their relentless support to make it possible.

## Author Contributions

**Conceptualization:** Anamul Hasan, Mohammad Shafiul Alam.

**Data curation:** Anamul Hasan, Md. Mobarok Hossain, Md Fahad Zamil, Afrida Tabassum Trina, Mohammad Sharif Hossain, Mustafizur Rahman, Mohammad Shafiul Alam.

**Formal analysis:** Anamul Hasan, Md. Mobarok Hossain, Md Fahad Zamil, Mustafizur Rahman, Mohammad Shafiul Alam.

**Investigation:** Anamul Hasan, Md Fahad Zamil, Afrida Tabassum Trina, Mohammad Sharif Hossain, Asifa Kumkum, Sajia Afreen, Dilruba Ahmed.

**Methodology:** Anamul Hasan, Md. Mobarok Hossain, Md Fahad Zamil, Mohammad Sharif Hossain.

**Project administration:** Mohammad Shafiul Alam.

**Resources:** Mohammad Shafiul Alam.

**Software:** Anamul Hasan, Md. Mobarok Hossain, Md Fahad Zamil, Mustafizur Rahman.

**Supervision:** Dilruba Ahmed, Mustafizur Rahman.

**Validation:** Mustafizur Rahman, Mohammad Shafiul Alam.

**Visualization:** Md. Mobarok Hossain, Mustafizur Rahman.

**Writing – original draft:** Anamul Hasan.

**Writing – review & editing:** Mohammad Shafiul Alam.

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
