## [Decision Letter · Decision Letter 0]

18 Nov 2024

PNTD-D-24-01446Concurrent Transmission of Zika Virus During the 2023 Dengue Outbreak in BangladeshPLOS Neglected Tropical DiseasesDear Dr. Alam, Thank you for submitting your manuscript to PLOS Neglected Tropical Diseases. After careful consideration, we feel that it has merit but does not fully meet PLOS Neglected Tropical Diseases's publication criteria as it currently stands. Therefore, we invite you to submit a revised version of the manuscript that addresses the points raised during the review process. Please submit your revised manuscript within 60 days Jan 17 2025 11:59PM. If you will need more time than this to complete your revisions, please reply to this message or contact the journal office at plosntds@plos.org. Please include the following items when submitting your revised manuscript: *
A rebuttal letter that responds to each point raised by the editor and reviewer(s). You should upload this letter as a separate file labeled 'Response to Reviewers'. This file does not need to include responses to any formatting updates and technical items listed in the 'Journal Requirements' section below. *
A marked-up copy of your manuscript that highlights changes made to the original version. You should upload this as a separate file labeled 'Revised Manuscript with Track Changes'. *
An unmarked version of your revised paper without tracked changes. You should upload this as a separate file labeled 'Manuscript'. If you would like to make changes to your financial disclosure, competing interests statement, or data availability statement, please make these updates within the submission form at the time of resubmission. Guidelines for resubmitting your figure files are available below the reviewer comments at the end of this letter. We look forward to receiving your revised manuscript. Kind regards,Andrea Morrison, Ph.D.Academic EditorPLOS Neglected Tropical Diseases Michael HolbrookSection EditorPLOS Neglected Tropical Diseases

Shaden Kamhawi

co-Editor-in-Chief

Paul Brindley

co-Editor-in-Chief

**Journal Requirements:**

1) Tables should not be uploaded as individual files. Please remove these files and include the Tables in your manuscript file as editable, cell-based objects. For more information about how to format tables, see our guidelines:

https://journals.plos.org/plosntds/s/tables 

2) In the online submission form, you indicated that "All Complete metadata set is available upon request from corresponding author." All PLOS journals now require all data underlying the findings described in their manuscript to be freely available to other researchers, either

1. In a public repository

2. Within the manuscript itself

3. Uploaded as supplementary information.

**Reviewers' Comments:** Reviewer's Responses to Questions

**Key Review Criteria Required for Acceptance?**

**Methods**

-Are the objectives of the study clearly articulated with a clear testable hypothesis stated?

-Is the study design appropriate to address the stated objectives?

-Is the population clearly described and appropriate for the hypothesis being tested?

-Is the sample size sufficient to ensure adequate power to address the hypothesis being tested?

-Were correct statistical analysis used to support conclusions?

-Are there concerns about ethical or regulatory requirements being met?

Reviewer #1: This is explorative study to find ZIKV infection on the dengue study

The objectives of the study are clearly articulated with a clear testable hypothesis stated

The population is clearly described and appropriate for the hypothesis being tested

the study design is appropriate to address the stated objectives

the sample size insufficient to ensure adequate power to address the hypothesis being tested

statistical analysis : NA

Yes they concern about ethical; or regulatory requirement

Reviewer #2: Hasan et al., in their manuscript entitled "Concurrent Transmission of Zika Virus During the 2023 Dengue Outbreak in Bangladesh" described 5 cases of Zika during a Dengue epidemic in Bangladesh, one of which was a co-infection of Zika virus and Dengue. The detection of five autochthonous cases of Zika in Bangladesh is a matter of concern considering that the number of countries reporting Zika is increasing worldwide.

In the introduction

line 63, according to ICTV the genus is Orthoflavivirus or use the family Flaviviridae.

Reviewer #3: Introduction

Line 86: The phrase "As far as we know" is too informal; please consider rephrasing it.

Methods

1. The title currently says "in Bangladesh," but your sample is only from Dhaka. Does this accurately represent Bangladesh? Consider revising it to "in Dhaka, Bangladesh."

2. Were clinical data collected for all participants? You mention in the Introduction (lines 96-97) "corresponding clinical and epidemiological risk factors." If so, it would be beneficial to present and analyze these data with the co-infection or ZIKV infection cases.

3. Line 141: You state that "detecting internal control is not essential." Does this mean you did not use any internal control in your method? If so, please explain why

**Results**

-Does the analysis presented match the analysis plan?

-Are the results clearly and completely presented?

-Are the figures (Tables, Images) of sufficient quality for clarity?

Reviewer #1: This is explorative study to find ZIKV infection on the dengue study, no analysis about Zika

The results is clearly presented

Figures are sufficients

Reviewer #2: In Results

a)line 179 - Rewrite the sentence better, as it implies that the patients had the same symptom, headache. Of the various symptoms associated with the disease, headache was the only one presented by the 5 cases?

b)line 185 in the paragraph - Findings from real-time RT-PCR analysis

Needs to make it clearer that only 3 samples were sequenced and justify that "maybe" was because of the high Ct values (low viral load)

c)line 196 - Please rewrite the sentence as it is giving another interpretation. It seems to the reader that this study identified 2 distinct lineages, when in fact the selected ZIKV dataset used sequences representing the two main lineages, Asian and African.

d)line 197- The figure needs to be better presented with this larger phylogenetic tree, plus a detailed description and location of the sequences for better visualization of the data.

Although the phylogenetic tree is well supported and represented, a better presentation of the figure with more enlarged detail should be done to better observe the comments described in the manuscript.

Reviewer #3: 1. From a total of 185 samples meeting the inclusion criteria, you performed RT-PCR on 152 samples. What happened to the remaining 67 samples? Please clarify this in the Methods or Results section. Including a flowchart would make this clearer.

2. Did you perform any immunological assays? It would be valuable to include immune response data to differentiate past versus current infections and illustrate the situation at the study site.

3. The finding of PCR, please add the specific RT-PCR you have done in the sentences, viral load of what virus?

**Conclusions**

-Are the conclusions supported by the data presented?

-Are the limitations of analysis clearly described?

-Do the authors discuss how these data can be helpful to advance our understanding of the topic under study?

-Is public health relevance addressed?

Reviewer #1: the conclusions are supported by the data presented

Yes the limitations of analysis are clearly described?

No

public health relevance is addressed?

Reviewer #2: 3)Discussion

a)line 217 - Please rewrite the sentence, as it is giving another interpretation. As described in the results, it seems to the reader that this study identified 2 distinct lineages, when in fact the selected ZIKV dataset used sequences representing the two main lineages, Asian and African

b)line 223 - It should be made clear in this paragraph that although the cases identified were only male and had an average age of 24 years, there is concern about the possibility of possible cases involving children born with congenital anomalies or who suffer premature death, which could have profound socioeconomic and psychosocial consequences in Bangladesh (4, 18)

c)line 236 - This information should appear before the paragraph on the line 223 – “As all suspected cases were male, the probability of male-to-female ZIKV transmission and subsequent pre- and peri-natal transmission from mother to descendants would be higher, potentially resulting in the birth of microcephalic neonates (45).”

Reviewer #3: If the conclusion in the abstract yes, but please consider to add the conclusion part after the discussion or add in the final paragraph of the discussion, that will be facilitate to add more information rather than only highlight.

**Editorial and Data Presentation Modifications?**

Reviewer #1: This manuscript needs major revision

Reviewer #2: Minor Revision

Reviewer #3: -

**Summary and General Comments**

Reviewer #1: The strength of the study if teh finding of ZIKV that may impact permanent deficit to the baby

Reviewer #2: (No Response)

Reviewer #3: This study was conducted in Dhaka, not throughout Bangladesh, so the sample may not be representative of the entire country. Please consider adding an analysis of clinical data along with the experiments performed, as well as including immunological assays, to make this paper more comprehensive.

PLOS authors have the option to publish the peer review history of their article (what does this mean?). If published, this will include your full peer review and any attached files.

Reviewer #1: No

Reviewer #2: No

Reviewer #3: No

---

## [Decision Letter · Decision Letter 1]

24 Jan 2025

Dear Dr. Alam,

We are pleased to inform you that your manuscript 'Concurrent Transmission of Zika Virus During the 2023 Dengue Outbreak in Dhaka, Bangladesh' has been provisionally accepted for publication in PLOS Neglected Tropical Diseases.

Best regards,

Andrea Morrison, Ph.D.

Academic Editor

Michael Holbrook

Section Editor

Shaden Kamhawi

co-Editor-in-Chief

Paul Brindley

co-Editor-in-Chief

Reviewer's Responses to Questions

**Key Review Criteria Required for Acceptance?**

**Methods**

-Are the objectives of the study clearly articulated with a clear testable hypothesis stated?

-Is the study design appropriate to address the stated objectives?

-Is the population clearly described and appropriate for the hypothesis being tested?

-Is the sample size sufficient to ensure adequate power to address the hypothesis being tested?

-Were correct statistical analysis used to support conclusions?

-Are there concerns about ethical or regulatory requirements being met?

Reviewer #1: The study objective are clearly stated. The study design is is address appropriately.

Reviewer #3: (No Response)

**Results**

-Does the analysis presented match the analysis plan?

-Are the results clearly and completely presented?

-Are the figures (Tables, Images) of sufficient quality for clarity?

Reviewer #1: The figures are clear

Reviewer #3: (No Response)

**Conclusions**

-Are the conclusions supported by the data presented?

-Are the limitations of analysis clearly described?

-Do the authors discuss how these data can be helpful to advance our understanding of the topic under study?

-Is public health relevance addressed?

Reviewer #1: Yes

Reviewer #3: (No Response)

**Editorial and Data Presentation Modifications?**

Reviewer #1: Accept

Reviewer #3: (No Response)

**Summary and General Comments**

Reviewer #1: The study has no novelty, but there is significance impact of this study in the Zika infection awareness.

Reviewer #3: (No Response)

PLOS authors have the option to publish the peer review history of their article (what does this mean?). If published, this will include your full peer review and any attached files.

Reviewer #1: No

Reviewer #3: No

---

## [Editor Report · Acceptance letter]

26 Jan 2025

Dear Dr. Alam,

We are delighted to inform you that your manuscript, "Concurrent Transmission of Zika Virus During the 2023 Dengue Outbreak in Dhaka, Bangladesh," has been formally accepted for publication in PLOS Neglected Tropical Diseases.

Best regards,

Shaden Kamhawi

co-Editor-in-Chief

Paul Brindley

co-Editor-in-Chief
